

# Fishing operation type recognition based on multi-branch convolutional neural network using trajectory data

Bohui Jiang[1,2] and Weifeng Zhou[1]

[1] East China Sea Fisheries Research Institute, Chinese Academy of Fishery Sciences, Shanghai, China
[2] Shanghai Ocean University, Shanghai, China

## ABSTRACT

Accurate identification of fishing vessel operations is vital for sustainable fishery management. Existing methods inadequately exploit spatiotemporal contextual information in vessel trajectories and fail to effectively fuse multimodal data. To address this, this study proposes a novel framework integrating Geohash-based geocoding with embedding techniques inspired by natural language processing to extract spatiotemporal features from trajectory sequences. We develop a multi-branch 1D convolutional neural network (MB-1dCNN) to minimize feature engineering dependency while enhancing operational-type recognition. Comparative experiments evaluate Geohash encoding lengths and network architectures (single-branch *vs.* multi-branch, fully-connected *vs.* 1D-CNN). Results indicate optimal Geohash encoding at length 5. The multi-branch structure significantly outperforms single-branch counterparts, and MB-1dCNN demonstrates superior performance over multi-branch model with fully connected layers (MB-FCNN), achieving additional gains in accuracy and F1-score. Key findings reveal: (1) 1D-CNN processing surpasses fully-connected networks in sequential feature extraction, (2) Multi-branch architectures enhance information fusion capabilities. The proposed MB-1dCNN establishes state-of-the-art performance for trajectory-based fishing operation recognition, offering valuable insights for spatial computing applications in maritime surveillance.

## INTRODUCTION

As terrestrial aquatic resources are increasingly exploited, oceans, which cover over 70% of the Earth's surface, are gaining attention. Consequently, the exploitation of marine resources has accelerated. Scientific progress has been made in fish stock planning and utilization (*FAO, 2020*; *Xu et al., 2021*). Various fishing methods impact marine fishery resources differently; particularly, excessive fishing activities severely damage specific marine populations (*Rousseau et al., 2019*). To conduct commercial activities, fishing vessel owners must apply for permits and register their operational types (*i.e.*, fishing methods) with competent authorities. However, fishing vessels often operate illegally,

Corresponding author
Weifeng Zhou,
zhou_wf@hotmail.com

violating local or international rules and harming fishery resources and the marine ecological environment (*Kassawmar et al., 2018*; *Rockström, Barron & Fox, 2002*). Traditionally, marine officers on patrol boats monitor fishing violations through boarding and inspecting fishing vessels. Yet, the large number of active fishing vessels, their unpredictable movements, and long operational periods make such monitoring challenging and expensive, requiring a substantial marine patrol workforce. Consequently, practical management of fishing vessel operations is limited. In this context, accurate identification of fishing operational types or activities using technology is critical for promoting better management of commercial fishing vessels and the sustainable development of fishery resources.

Vessel tracking provides location information, increasing transparency in fishing activities (*Orofino et al., 2023*). Vessel monitoring systems (VMSs) and automatic identification systems (AISs) have been applied in fishing vessel management (*Mesquita et al., 2024*; *Iacarella et al., 2023*). However, existing methods fail to fully utilize the spatiotemporal contextual information contained in the fishing-vessel trajectories, and poorly fuse multiple information of inputs for fishing-vessel operational-type identification. Considering the different ways in which input features are processed, the research motivation of the study is to reduce the dependence of previous methods on feature engineering and exploiting extra spatiotemporal contextual information to build a new and relatively good acceptable neural network model to do the recognition of fishing vessel operation-types. The approach proposed in this research utilizes the spatioltemporal contextual feature extraction of trajectory sequences. Additionally, it contributes to the extraction and fusion of multiple-input feature information, which may be advantageous for applications of spatial computation and analysis.

## RELATED WORK

Fishing vessel trajectory data from VMSs or AISs, in combination with expert knowledge and machine learning techniques, can facilitate the identification of various fishing operational types or activities. VMSs record real-time data for fishing vessel trajectory points, including longitude, latitude, speed, heading, and timestamp. This enables the rapid acquisition of positional data over large areas (*Walker & Bez, 2010*; *Vermard et al., 2010*; *Zhou et al., 2015*), making VMSs crucial for monitoring and managing fishing efforts. The speed and fishing activity of a single vessel vary across different statuses, such as sailing and operation. *Natale et al. (2015)* and *Zheng et al. (2016)* conducted analyses of VMS trajectory data to identify patterns. Similarly, *Gao et al. (2020)* and *Huang et al. (2019)* extracted features from VMS trajectory data and employed classical machine learning algorithms—like XGBoost, support vector machines, random forests, and k-nearest neighbors—to recognize fishing activity patterns. *Li et al. (2021b)* applied a Gaussian mixture model to mine characteristic thresholds based on the statistical frequency distribution of fishing vessel operating speeds. *Yang et al. (2022)* constructed 17-dimensional vessel characteristics related to spatial, temporal, and behavioral parameters and used a three- layer bidirectional long short-term memory (BiLSTM) network to classify fishing and non-fishing vessels. *Li et al. (2021a)* extracted a
42-dimensional feature vector based on a LightGBM-BiLSTM model to effectively identify fishing activity patterns. *Natale et al. (2015)* developed a Gaussian mixture model to identify fishing behavior from AIS data. By setting speed distribution thresholds, they categorized trajectories into navigation, fishing, and anchoring behaviors, thereby distinguishing fishing from non-fishing states. *Zhang et al. (2021)* analyzed fishing vessel behavior using vessel speed, sailing time, trajectory, and fishing effort. This allowed for the direct identification of fish stock locations and the statistical analysis of fishing output and resource distribution. *Ashrafi, Tessem & Enberg (2023)* utilized AIS data along with several deep learning architectures to detect likely unreported fishing activities, with a particular focus on bottom trawlers.

All these approaches rely on statistical analyses of the fishing industry's state to obtain parameters such as possible thresholds. Furthermore, feature engineering is performed to identify different fishing operational types. Over 10 features are constructed from trajectory data. *Vermard et al. (2010)* proposed a Bayesian hierarchical model based on hidden Markov processes. Using speed and movement transfer as parameters, this model predicts vessel states of different behaviors, including fishing, sailing, and docking. *Feng et al. (2019)* used VMS data and BP neural networks, with direction angle and speed changes as inputs, to effectively identify fishing behaviors. *Gao et al. (2020)* developed a feature fusion algorithm by extracting features from VMS data to determine the operation type.

However, these methods are highly dependent on thresholds or feature engineering. The accuracy of pattern identification is determined by reasonable threshold settings and valid feature extraction. Determining these thresholds or features requires a certain level of expert knowledge, as parameters vary with different fishing activity patterns. Consequently, these methods have limitations that hinder their generalized application.

Compared to classical machine learning, which requires the manual construction of multiple features, deep learning offers the advantage of end-to-end learning (*LeCun, Bengio & Hinton, 2015*). Deep-learning models can automatically learn features from datasets using network structures, thereby reducing reliance on expert knowledge (*Janiesch, Zschech & Heinrich, 2021*). As a result, deep learning is increasingly being applied to various tasks. Notably, *Kim & Lee (2020)* used convolutional neural network (CNN) models trained on vessel trajectory data to distinguish fishing from non-fishing vessels. Additionally, *Tang et al. (2020)* generated a track map of vessel trajectory data and classified gillnet and trawl-net types using the Visual Geometry Group (VGG)-16 model. All these methods demonstrated plausible performance in fishing-activity pattern identification.

A trajectory sequence consists of continuous track points in time and space. In addition to the spatial and temporal information of the track points, the trajectory sequence contains spatiotemporal contextual information (*Garani & Adam, 2020*). However, the methods mentioned above do not fully utilize this contextual information when classifying fishing activity patterns. If the geographic location of a track point can be converted into a literal representation, then the trajectory sequence can be expressed as a piece of text, with a certain point regarded as a word within the text. This allows for the extraction of

contextual information to facilitate geographical analysis and achieve spatial "semantic" information. Consequently, techniques used to extract contextual information in natural language word processing can be applied to process trajectory sequences. Previously, *Yin et al. (2019)* used Universal Transverse Mercator coordinates to discretize the Earth's surface. They then applied geographical units of different granularities to perform GPS encoding and finally trained a neural network to learn the spatial contextual information of global geographical coordinates. *Mai et al. (2020)* obtained the absolute locations and spatial relationships of encoded locations using the Space2Vec representation learning model. *Tian et al. (2022)* employed the GCN aided Location2Vec (GCN-L2V) model to learn the contextual semantic relationships between locations. Therefore, the representation of trajectory sequences can also be applied to the identification of fishing vessel operation types.

In summary, existing methods fail to fully utilize the spatiotemporal contextual information contained in fishing-vessel trajectories and exhibit poor fusion of multiple input information for fishing-vessel operational-type identification.

## MATERIALS AND METHODS

In this study, a new adaptive recognition method based on a convolutional neural network was designed employing 1D convolutions and a multi-branch structure. The proposed algorithm, called a multi-branch convolutional neural network (MB-1dCNN), realizes operational type recognition through the enhancement processing of fishing vessel trajectories. This method was applied in experiments to identify fishing-vessel activity patterns for three fishing-vessel operational types: purse seine, trawl net, and gillnet fishing. And in the comparative experiments, the spatial contextual information, the influence of Geohash encoding length and different convolutional neural network structures, such as fully connected network and 1D convolutions, as well as single-branch and multi-branch, on the recognition effect is explored. A single-branch neural network, primarily composed of fully connected layers (SB-FCNN), serves as a benchmark.

This section provides an overview of the Geohash algorithm, embedding, and fully connected neural networks (FCNN) and 1D-CNNs. The structure of single- and multi-branch are also explained, as these components form the basis of the proposed and comparative models explored in this study. In the subsequent section, we compare the results of the MB-1dCNN model with those of a single-branch model with fully connected layers (SB-FCNN), a multi-branch model with fully connected layers (MB-FCNN), and a single-branch structure with 1D convolutions (SB-1dCNN). All architectures can be found in Fig. 1. Figure 1D represents the model structure proposed and recommended in this article, while Figs. 1A, 1B, and 1C respectively represent the model structures of SB-FCNN, SB-1dCNN, and MB-FCNN. The constituent modules of the models are shown in Table 1. Additionally, the datasets and evaluation metrics used in this study are also presented in this section.

The contributions of this study are summarized as follows:

(1) The Geohash location encoding algorithm converts 2D geographic coordinates of the trajectory points into compact 1D alphanumeric strings, combined with the

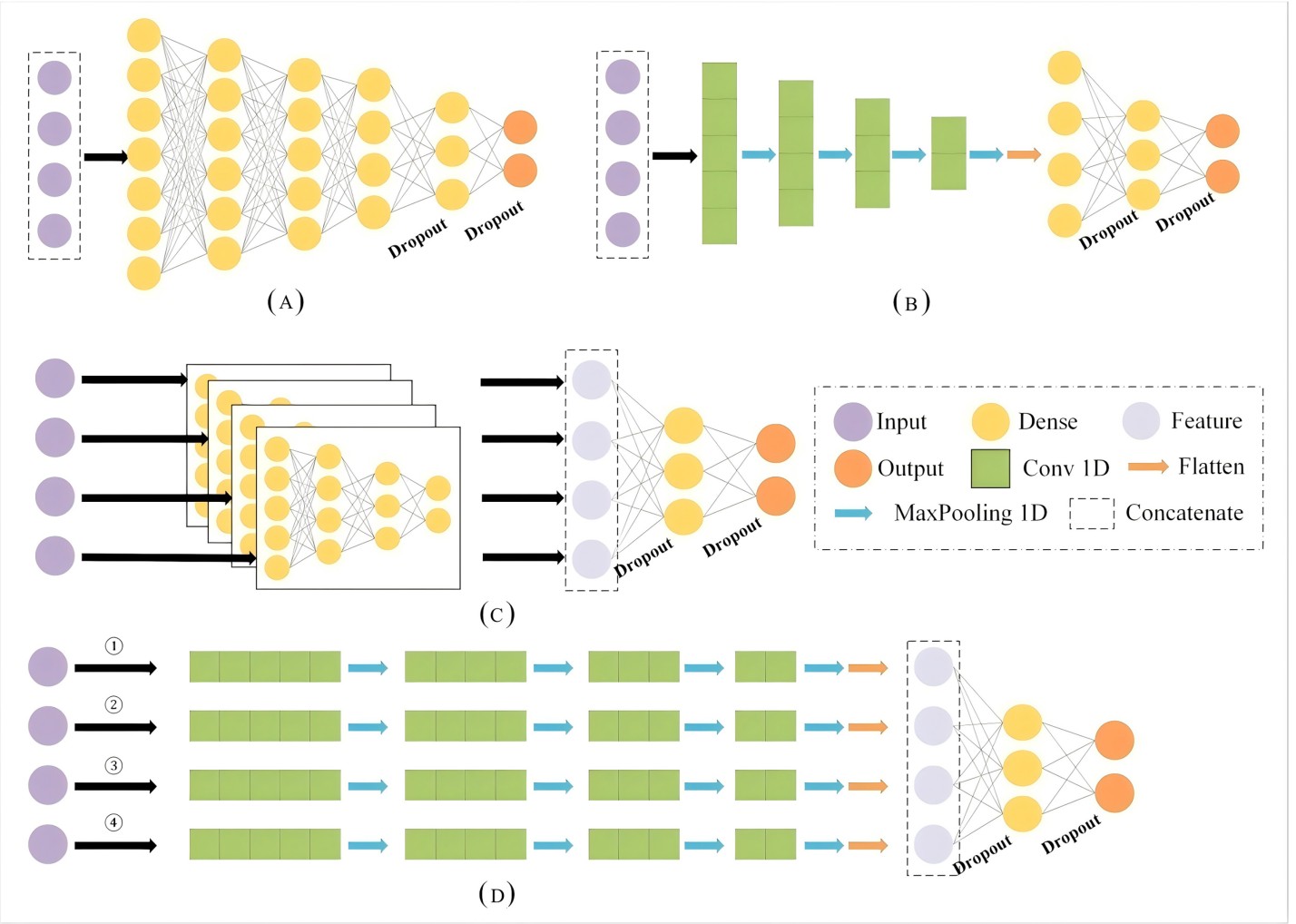

**Figure 1 The architectures of different neural networks.** (A) SB-FCNN (B) SB-1dCNN (C) MB-FCNN (D) MB-1dCNN.

embedding technique to capture the spatiotemporal contextual information from trajectory sequences.

(2) A multi-branch convolutional neural network architecture utilizing 1D convolutions for fishing vessel operational-type recognition.

(3) Experimental validation demonstrating that Geohash encoding with length 5 achieves optimal recognition accuracy.

(4) Empirical evidence showing that 1D convolutional architectures outperform fully connected networks, and multi-branch designs surpass single-branch counterparts in this task.

## Geohash

The Geohash algorithm, which was proposed by Gustavo Niemeyer in 2008, is a geocoding method in the public domain (*Yin & Chen, 2022*; *Zhou et al., 2021*). What's particularly

**Table 1 The constituent modules of the neural networks.**

|  | SB-FCNN | MB-FCNN | SB-1dCNN | MB-1dCNN |
|---|---|---|---|---|
| Input | 800 | 800 | 800 | 800 |
| Embedding | 20,000, 50 | 20,000, 50 | 20,000, 50 | 20,000, 50 |
| GlobalAvgPool1D | √ | √ | × | × |
| Concatenate | 4 | × | 4 | × |
| Dense | 128 | 128 | × | × |
| Dense | 64 | 64 | × | × |
| Dense | 32 | 32 | × | × |
| Dense | 16 | 16 | × | × |
| Conv1D | × | × | 4, 5 | 4, 5 |
| MaxPooling1D | × | × | 2 | 2 |
| Conv1D | × | × | 8, 5 | 8, 5 |
| MaxPooling1D | × | × | 2 | 2 |
| Conv1D | × | × | 16, 5 | 16, 5 |
| MaxPooling1D | × | × | 2 | 2 |
| Conv1D | × | × | 32, 5 | 32, 5 |
| MaxPooling1D | × | × | 2 | 2 |
| Flatten | × | × | √ | √ |
| Concatenate | × | 4 | × | 4 |
| Dropout | 0.5 | 0.5 | 0.5 | 0.5 |
| Dense | 8 | 8 | 128 | 128 |
| Dropout | 0.5 | 0.5 | 0.5 | 0.5 |
| Output | 3 | 3 | 3 | 3 |

elegant about this algorithm is that it follows the pattern of "left is 0, right is 1; bottom is 0, top is 1", which is used to represent the area after subdivision, achieved by continuously and alternately dichotomizing the global longitude [−180°, 180°] [−180°, 180°] and latitude [−90°, 90°] ranges. In the Geohash approach, the corresponding binary value is recorded until the encoding length reaches the required accuracy. The longitude and latitude are then rearranged into a new binary string. Finally, the corresponding Geohash string is obtained using Base32, which encodes every five binary digits (*Suwardi et al., 2015*; *Li et al., 2020*). The alphabetically ordered Geohashes trace out a Z-order curve. As shown in Fig. 2, at the first level of Geohash encoding, different global ranges can be represented by separate 32 American Standard Code for Information Interchange (ASCII) characters. If the positions of the characters are connected with curves in the usual order, the curves appear Z-shaped. This is so called as a Z-order filling curve.

Thus, the Geohash algorithm converts the 2D longitude and latitude coordinates of a point into a 1D string to represent the corresponding geographical location (*Zhou et al., 2022*, *2020*). The length of the Geohash string depends on the encoding accuracy; a longer string provides higher accuracy, and vice versa. A Geohash is generally encoded at Levels 1–12, corresponding to a grid range from $5,000 \times 5,000$ km$^2$ to $0.000000372 \times 0.000000186$ km$^2$ like Table 2. Each Geohash code represents a grid region on Earth and is

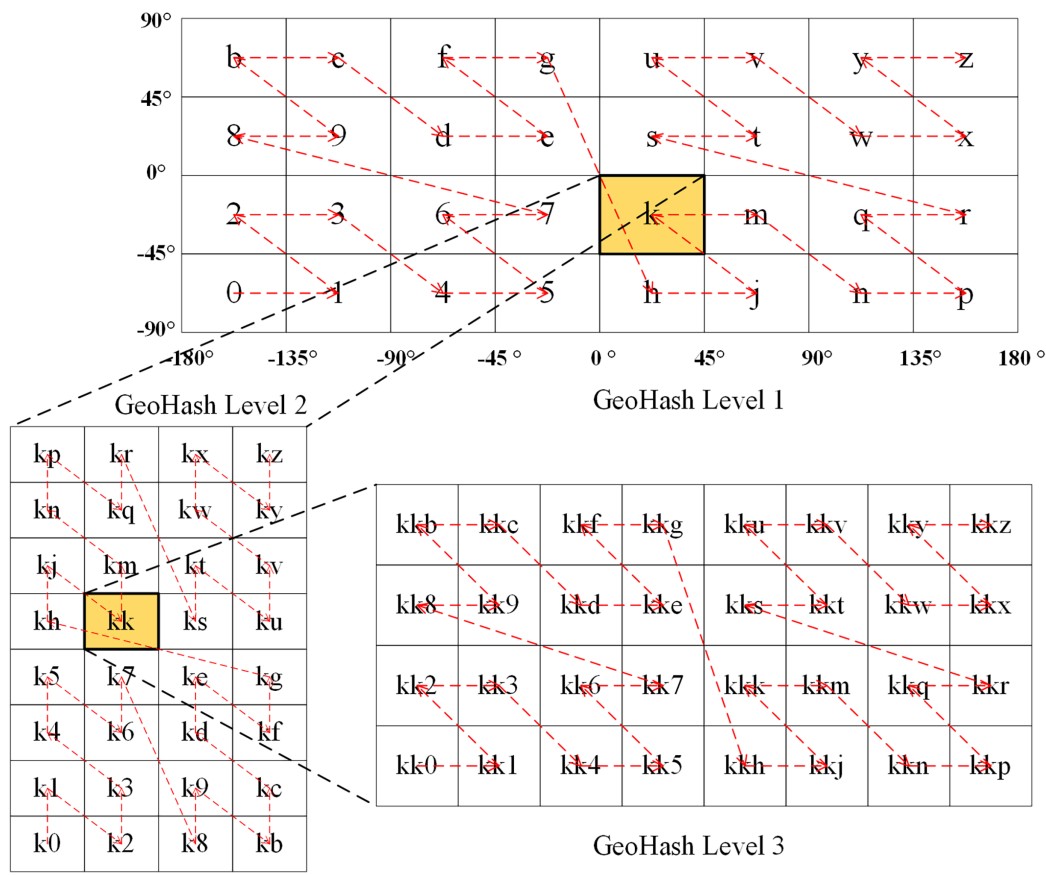

**Figure 2 Illustration of the level 1–3 of Geohash geocoding.**

**Table 2 Precision correspondence between the GeoHash encoding level and geospatial size.**

| Geohash level | Precision in latitude | Precision in longitude | Precision expressed in kilometers | Grid width in kilometers | Grid height in kilometers |
|---|---|---|---|---|---|
| 1 | ±23 | ±23 | ±2,500 | 5,000 | 5,000 |
| 2 | ±2.8 | ±5.6 | ±630 | 1,250 | 625 |
| 3 | ±0.7 | ±0.7 | ±78 | 156 | 156 |
| 4 | ±0.087 | ±0.18 | ±20 | 39.1 | 19.5 |
| 5 | ±0.022 | ±0.022 | ±2.4 | 4.89 | 4.89 |
| 6 | ±0.0027 | ±0.0055 | ±0.61 | 1.22 | 0.61 |
| 7 | ±0.00068 | ±0.00068 | ±0.076 | 0.153 | 0.153 |
| 8 | ±0.000086 | ±0.000172 | ±0.01911 | 0.0382 | 0.0191 |
| 9 | ±0.000021 | ±0.000021 | ±0.00478 | 0.00477 | 0.00477 |
| 10 | ±0.00000268 | ±0.00000536 | ±0.0005971 | 0.00119 | 0.000596 |
| 11 | ±0.00000067 | ±0.00000067 | ±0.0001492 | 0.00000149 | 0.00000149 |
| 12 | ±0.00000008 | ±0.00000017 | ±0.0000186 | 0.000000372 | 0.000000186 |

the same for all coordinate points within that grid. As the Geohash string length increases, the extent of the grid it represents decreases (*Irshaid et al., 2021*; *Jin et al., 2019*; *Guo et al., 2019*).

## Embedding

Embedding techniques were initially used in natural language processing tasks to convert string text into numeric vectors, primarily focusing on representation of words and documents (*Mikolov et al., 2013*; *Zhao et al., 2020*; *Sonbol, Rebdawi & Ghneim, 2022*). This method can be used to convert a word into a vector; that is, the text is converted into a fixed-length vector of consecutive real numbers. Words in a vocabulary are mapped to a potential vector space that summarizes their syntactic and semantic information. The basic assumption of the embedding technique is that words with the same meaning in a text should have similar representations and, thus, embedding vectors that are closer in the vector space are more similar in textual meaning. Therefore, the embedding representation can reveal the background information, that is the textual semantic information (*Babić, Martinčić-Ipšić & Meštrović, 2020*; *Wang, Nulty & Lillis, 2021*). Current implementations of embedding methods for natural language processing include Word2vec, a static embedding method based on local semantic information (*Mikolov et al., 2013*), and Embeddings from Language Models (ELMo), a method based on dynamic semantic information (*Wang et al., 2020*).

In this study, embedding is implemented using the TensorFlow embedding layer. We do not directly use functions from the Gensim library, which is a popularly used library for Word2vec algorithm's implementations. The embedding layer in TensorFlow being used to process text sequences can be considered an implement of the Word2vec algorithm. An embedding layer is primarily used to encode a sentence to produce a distributed representation of a word or character. After a sentence passes through this layer, the word vector of each word or character can be obtained, where $k$ denotes the dimensions of the word vector and $X_i$ denotes the $i$-th word. A sentence of length $n$ can then be expressed as

$$X_{1:n} = X_1 \oplus X_2 \cdots \oplus X_n \tag{1}$$

where $\oplus$ denotes the concatenation operator and $X_{i:i+j}$ denotes the matrix of feature vectors formed by the word vectors $X_i, X_{i+1} \ldots, X_{i:i+j}$.

In the proposed model, the convolutional, pooling, and fully connected layers for the classification task are positioned behind the embedding layer. This design enables the embedding layer to process semantic information for weight adjustment considering both the contextual information of the word and the classification task, thereby improving the semantic information extraction performance. Then the embedding layer is realized using the function tf.keras.layers.Embedding in TensorFlow to establish the spatiotemporal contextual features of the trajectory segment by an end-to-end method. Table 1 lists the parameters used in this study.

## Neural network model construction and comparison
### Fully connected neural networks and one-dimensional convolutional neural networks

An FCNN is a classical artificial neural network structure that comprises multiple neurons connected according to a hierarchical structure (*Liu et al., 2018*). FCNNs typically comprise an input layer, several hidden layers, and an output layer. The input layer receives raw data as input, the hidden layers extract higher-order data features, and the output layer outputs the corresponding results based on the task requirements. Each hidden and output layer contains multiple neurons, and the number of neurons and layers in each hidden layer can be adjusted according to the task complexity and data characteristics. During the training process, the FCNN continuously adjusts the connection weights through a back-propagation algorithm and optimizer so that the network output is as close as possible to the real value and the loss function is minimized. By using a large number of training samples and iterative optimization, the FCNN can learn the complex patterns and laws of the supplied data. Further, FCNNs have strong generalization ability (*Liu et al., 2019*; *Lu et al., 2021*). An FCNN can be expressed as

$$\hat{y} = f\left(\sum w_i x_i + b\right) \tag{2}$$

where $x_i$ denotes the input, $w_i$ denotes the weight, $b$ denotes the bias, $f$ denotes the activation function, and $\hat{y}$ denotes the output.

A CNN is a model structure that gradually learns data features from low- to high-level patterns (*Yamashita et al., 2018*). The basic CNN structure comprises input, convolutional, pooling, fully connected, and output layers. The first stage involves the continuous extraction of features from the input data through alternations between the convolutional and pooling layers, as well as data dimensionality reduction through the use of a fully connected layer when approaching the output layer (*Zhou, Jin & Dong, 2017*). A one-dimensional convolution (1D-CNN) is typically used to process 1D sequential data.

In a 1D-CNN model structure, 1D data are used as input and the data features are continuously extracted using 1D convolution and pooling (*Qazi, Almorjan & Zia, 2022*). In the convolutional layer, the kernel performs a convolution operation on the feature-vector output of the previous layer. The output feature vector is then constructed using a nonlinear activation function. The output of each layer is the result of the convolution of multiple-input features. The mathematical model can be expressed as

$$X_j^l = F\left(\sum_{i \in M_j} X_i^{l-1} \times k_{ij}^l + b_j^l\right) \tag{3}$$

where $M_j$ represents the set of input features or receptive field for the $j$-th output neuron in a convolutional layer; $l$ denotes the $l^{\text{th}}$ network layer; $k$ denotes the convolution kernel; $b$ denotes the network bias; $X_i^l$ and $X_i^{l-1}$ denote the $l^{\text{th}}$-layer output and input, respectively; and $F(\cdot)$ denotes the activation function.

### Single- and multi-branch structures

Single- and multi-branch strategies are two alternative model-building strategies. For a model with a single-branch structure, multiple factors are stitched together and fed into the model. When a multi-branch structure is employed, factor information is extracted using a separate substructure; the extracted information is then combined and fed into the subsequent network for learning.

In this study, we developed a new MB-1dCNN model based on the Geohash algorithm, word embedding, and CNN deep learning methods. The model was designed to identify the activity patterns of commercial fishing vessels. The model structure is illustrated in Fig. 1D. Additionally, we constructed the SB-FCNN (Fig. 1A), MB-FCNN (Fig. 1C), and SB-1dCNN (Fig. 1B) models.

The MB-FCNN model can be expressed as

$$X_t = C(\hat{y}_1, \ldots, \hat{y}_n) \tag{4}$$

where $\hat{y}_1$ and $\hat{y}_n$ denote the outputs of the first and $n^{\text{th}}$ layers, respectively; $C(\cdot)$ denotes multiple-value splicing; and $X_t$ denotes the layer after splicing.

The proposed MB-1dCNN model extracts the features of each input factor separately through convolution and pooling and then combines the extracted information in a fully connected layer before output. Compared to a CNN, into which all factors are input, the MB-1dCNN can extract the features of each factor more efficiently. The mathematical model can be expressed as

$$X_t = C\left(\underbrace{FA\big(MP(X_i^l)\big)}_{X_j^l}, \ldots\right) \tag{5}$$

where $MP(\cdot)$ denotes the pooling operation, $FA(\cdot)$ denotes the flattening function, $C(\cdot)$ denotes the splicing of multiple values, and $X_t$ denotes the layer after splicing.

Overall, the SB-FCNN model comprises an input layer, five dense layers, two dropout layers, and an output layer; the MB-FCNN model comprises an input layer, four dense layers for separate extraction of each input feature, two dropout layers, one dense layer for extraction of the combined feature information, and an output layer; the SB-1dCNN model comprises an input layer, four Conv1D layers, MaxPooling1D layer modules, two dropout layers, a dense layer, and an output layer; and the MB-1dCNN model comprises an input layer, four Conv1D layers and MaxPooling1D layer modules for separate extraction of each input feature, two dropout layers, a dense layer for extraction of the combined feature information, and an output layer. The parameters of the four models are listed in Table 1.

All of these models have four inputs: the location data through embedding processing after Geohash geocoding transformation, the embedding time data, speed and heading. Since the representation method of words in natural language processing is referenced, the vector obtained after embedding processing after Geohash transformation can be considered to contain the spatiotemporal contextual information of the trajectory. Except

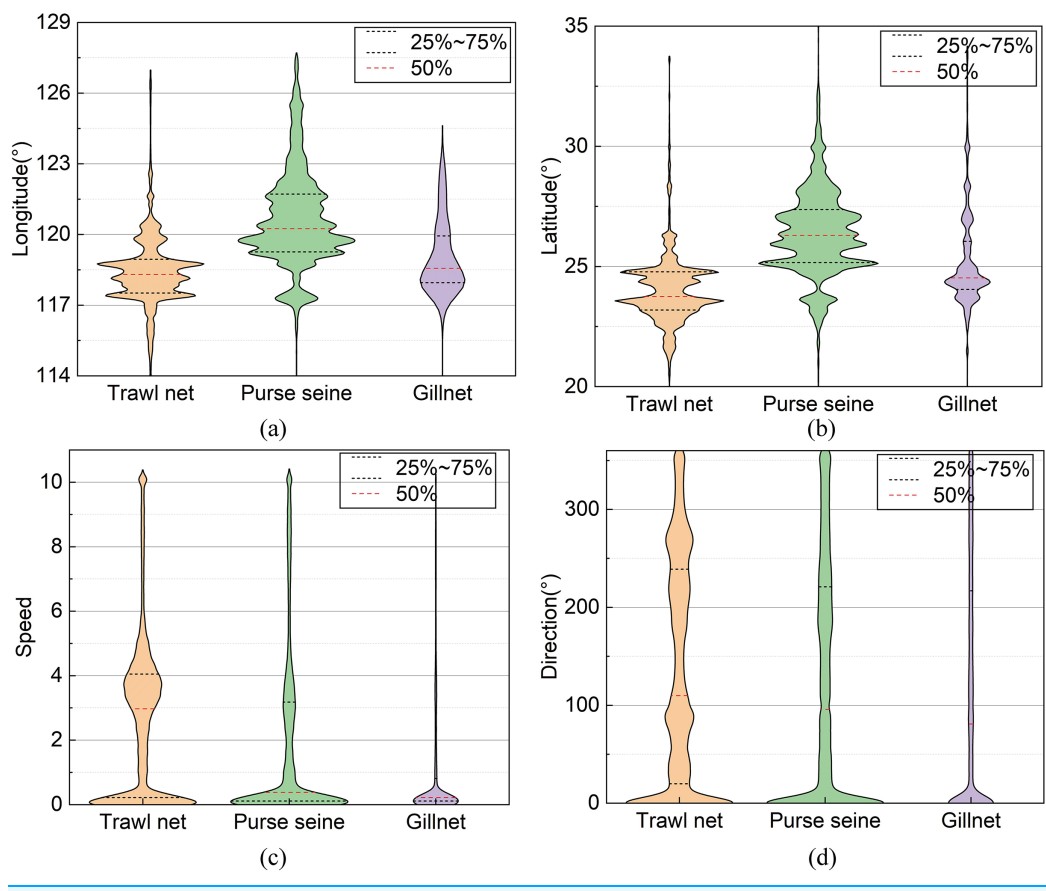

**Figure 3 Distribution of field values in the datasets.**

that the activation function used in the last layer of the models is SoftMax, the activation function used in other layers is rectified liner unit (ReLu) activation function. The Adam optimizer was implemented with a learning rate of 0.001; CategoricalCrossentropy was used as the loss function. The GPU was a GeForce RTX 2080 Ti run on a Windows operating system. The MB-1dCNN model was built using Python 3.7 and TensorFlow 2.4.1.

## Datasets

Experimental data were obtained from the 2020 Digital China Innovation Competition (https://tianchi.aliyun.com/competition/entrance/231768/introduction), which provided trajectory data for three fishing-vessel operational types: trawl net, purse seine, and gillnet fishing. In these datasets, a trajectory segment comprises several track points of the vessel at sea based on a unique vessel ID, along with the fishing type corresponding to that track segment.

In this study, these data were preprocessed to remove operational track segments having fewer than five track points and heading directions outside the range of 0–360°. The final data volume was 8,163. The data fields comprised the fishing vessel ID, longitude, latitude, speed, heading, timestamp, and registered fishing method or operational type. The

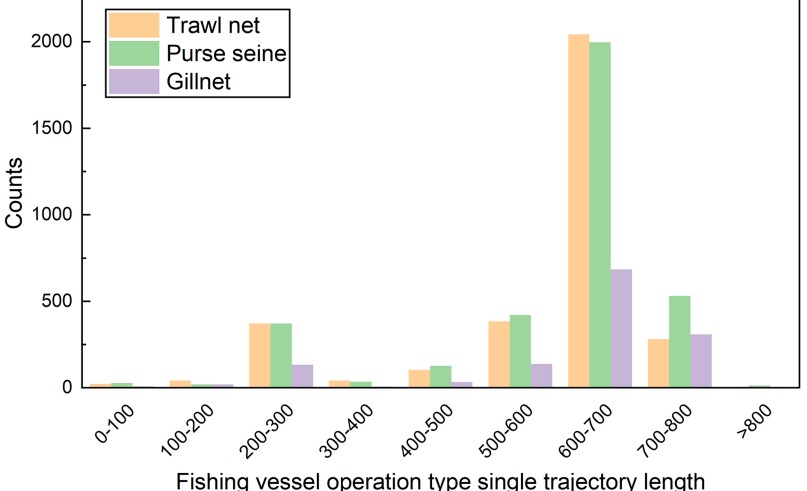

**Figure 4 Length distribution of trajectory segments.**

**Table 3 The amount of data in the datasets for different fishing operation types.**

|  | Purse seine | Trawl net | Gillnet | Total data |
|---|---|---|---|---|
| Train | 2,479 | 2,302 | 933 | 5,714 |
| Validation | 708 | 658 | 266 | 1,632 |
| Test | 354 | 329 | 134 | 817 |
| Total data | 3,541 | 3,289 | 1,333 | 8,163 |

Note:
Accuracy, Precision, Recall, and F1 values in the table are the weighted averages of the three operation types. The weight is the proportion of each operation type to the total samples.

longitude, latitude, speed, and direction ranges were [0°, 127.803°], [0°, 44.824°], [0, 11] knots, and [0°, 360°], respectively. The distribution of each feature value domain in the dataset is shown in Fig. 3. The length distribution of the single operational trajectories and the data volumes for the different fishing behaviors are shown in Fig. 4. The track-point sequence length of a single-vessel trajectory segment ranged from 21 to 5,841, with an average length of ~598. From Fig. 4, it is evident that the majority of trajectory sequences have a length of less than 800. Consequently, the trajectory length for the model's input has been determined to be 800. The dataset was divided into training, validation, and test datasets at a ratio of 7:2:1. The details of the dataset division are presented in Table 3.

The spatiotemporal contextual features of the fishing vessel trajectories were extracted from the original dataset and used as inputs for the recognition model. The model input features included the tensors of the spatiotemporal contextual features of the fishing vessel trajectories, times, speeds, and headings. The spatiotemporal context tensor of the fishing-vessel trajectories was obtained by encoding the longitude and latitude data using the Geohash method and performing extraction by embedding using natural language processing. The time feature was obtained by converting the "Month–Day" data to a

character string; this information was then represented by a vector obtained by embedding of nature language processing. The maximum-minimum normalisation method (*Bhanja & Das, 2018*) is used to preprocess the data on speed and direction of heading, so as with the non-geohash transformation geolocation data for the comparison.

## Evaluation metrics

Learner or model generalizability is a performance metric that requires valid and feasible experimental estimation methods and evaluation criteria. The performance metrics reflect the task requirements, and different metrics can often yield different results when the capabilities of different models are compared.

Following the construction of the complete classifier, we used a series of evaluation metrics to determine its effectiveness. *Accuracy* is one of the most commonly used performance measures in traditional classifier evaluation metrics, calculated as the ratio of correctly classified samples to the total number of samples. Typically, this metric can be applied to standard balanced data having a positive-to-negative sample ratio of 1:1. However, the number of samples is usually inconsistent between categories. In such cases, *Accuracy* should not be the only metric used to evaluate the classifier. For example, for a sample with a positive-to-negative sample ratio of 999:1, the model *Accuracy* can reach 99.9% even if the classifier predicts only the negative sample as positive. However, only the negative-sample information must be considered.

Consequently, another evaluation metric *F1* is introduced, for which a more significant value indicates better model classification. *F1* takes into account both *Recall* and *Precision*. *Recall* denotes the proportion of all positive samples predicted to be correct in the classification process, as well as *Precision* denotes the proportion of samples predicted to be in the positive class by the model during classification, calculated as the percentage of the correct predictions made by the model.

To reduce the resulting instability caused by the randomization of the neural network parameters, for both *Accuracy* and *F1*, we used the mean ± standard deviation values of the maximum and minimum values obtained after 12 training sessions under the same conditions for the same model.

## RESULTS AND DISCUSSION

### Influence of spatiotemporal contextual information on the classification accuracy of fishing vessel operation types

To investigate the impact of spatiotemporal contextual information on the classification accuracy of fishing vessel operation types, the comparison experiment using two different sets of trajectories data was designed. One set utilized latitude and longitude data directly for model training and prediction. The other set incorporated the spatiotemporal context information of the trajectories for the same purpose. The spatial contextual feature extraction of the trajectory were extracted using the GeoHash and Embedding techniques. Subsequently, the MB-1dCNN model was employed for training and prediction, achieving accuracy and F-score of 86.93% and 83.84% respectively on the test set.

**Table 4 Spatiotemporal contextual information effect on the fishing operation type recognition.**

| Features | Accuracy | F-score |
|---|---|---|
| Lon&Lat | 0.7280 ± 0.0045 | 0.6148 ± 0.0110 |
| GH—EM (Lon&Lat) | 0.8693 ± 0.0040 | 0.8384 ± 0.0053 |

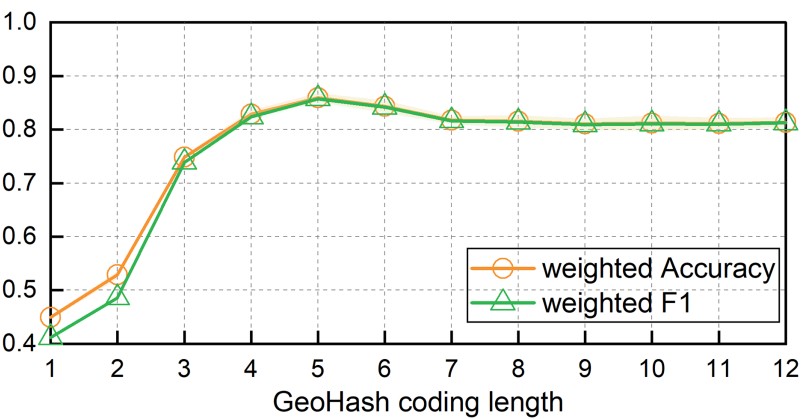

**Figure 5 Model results of MB-1dCNN based on different lengths of geohash encoding.**

As shown in Table 4, the comparison of these two experimental sets reveals that the classification accuracy and F-score improved by 14.13% and 22.36% respectively when using the Embedding method based on GeoHash transformation of vessel trajectories, compared to the model without spatiotemporal contextual information. This demonstrates that incorporating spatial contextual information from trajectories into the fishing vessel operation type classification model (MB-1dCNN) is highly effective in enhancing classification accuracy. Consequently, the spatiotemporal contextual information of trajectories promises to play a significant role in identifying fishing vessel types.

## Impact of different Geohash lengths on results

To address the issue of identifying fishing vessel operation types, we examined the effect of varying Geohash lengths on the MB-1dCNN model's ability to identify operation types using the spatiotemporal contextual features of fishing vessel trajectories by comparing the accuracy. Figure 5 illustrates a plot of the change in recognition accuracy with the Geohash length. As the Geohash length increased, both the weighted *Accuracy* and weighted *F1* scores of the model exhibited an upward trend followed by a decline. The highest classification accuracy was achieved at a Geohash length of 5; the weighted *Accuracy* was 0.8595 ± 0.0082 and weighted *F1* being 0.8573 ± 0.0085. The corresponding spatial extent was approximately 4.89 × 4.89 km$^2$.

As previously noted, each Geohash code represents a grid region on Earth, with the code length denoting the grid-region size. Figure 6 presents results for the gillnet, trawl net, and purse seine fishing operations in columns from left to right, respectively. The first row of Fig. 6 shows the original fishing vessel trajectories for each of these operations. The second

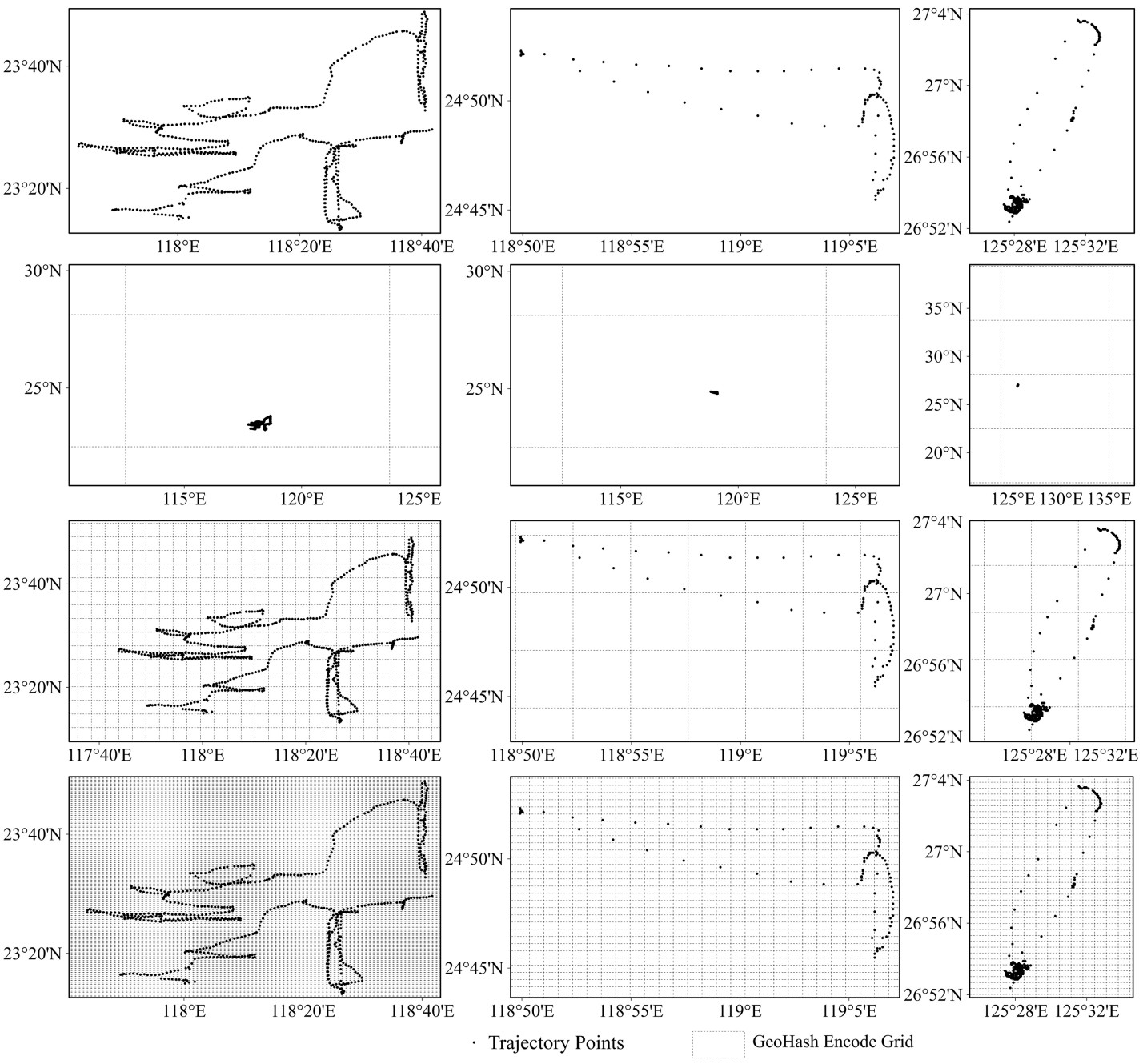

**Figure 6 Trajectories of different fishing types overlaid with the grids represented by the different geohash encoding lengths.** From left to right: gillnet, trawl net, and purse seine. From top to bottom: original trajectory; overlay of original trajectory points with geohash grid when the encoding length is 2, 5, and 6 respectively.

to fourth rows (from top to bottom) present overlays of the Geohash geographic grids on the vessel trajectories when the Geohash lengths were 2, 5, and 6, respectively. When the Geohash length was too short (as seen in the second row of Fig. 6, with length = 2), the geospatial region was larger, with most of the trajectory points enrolled in one region. This led to information loss under the Geohash coding, thereby reducing the model

**Table 5 Evaluation metrics of different neural networks for the recognition effect of fishing operations type.**

| Model | Accuracy | Precision | Recall | F1 |
|---|---|---|---|---|
| SB-FCNN | 0.7841 ± 0.0334 | 0.7178 ± 0.0907 | 0.7841 ± 0.0334 | 0.7362 ± 0.0601 |
| MB-FCNN | 0.8365 ± 0.0341 | 0.8187 ± 0.0714 | 0.8365 ± 0.0341 | 0.8197 ± 0.0580 |
| SB-CNN | 0.8660 ± 0.0037 | 0.8648 ± 0.0041 | 0.8660 ± 0.0037 | 0.8636 ± 0.0038 |
| MB-CNN | 0.8671 ± 0.0058 | 0.8669 ± 0.0059 | 0.8671 ± 0.0058 | 0.8651 ± 0.0058 |

classification accuracy. As the Geohash length gradually increased, the region size became more reasonable, and the specific fishing behavioral characteristics of the different fishing vessel types became evident after sequence encoding. Consequently, model classification accuracy improved (see the third row of Fig. 6, with length = 5). However, when the Geohash length was excessively long, the region division granularity became too fine, introducing noise and reducing model classification accuracy (see the fourth row of Fig. 6, with length = 6).

## Impact of different network structures on the model

We examine the recognition accuracies of the different models for the three fishing vessel operational types under investigation in this study. As noted previously, we utilized the trajectory-point latitude and longitude to construct the spatiotemporal contextual feature. It was then used as input feature for the models, alongside embedded-time, speed, and heading. As depicted in Table 5, the SB-1dCNN model achieved a weighted average *Accuracy* of 0.8660 ± 0.0037 and *F1* score of 0.8636 ± 0.0038. The MB-1dCNN model showed slightly higher values, with a weighted average *Accuracy* of 0.8671 ± 0.0058 and *F1* score of 0.8651 ± 0.0058. Both models demonstrated superior performance to the SB-FCNN and MB-FCNN models.

The SB-1dCNN and MB-1dCNN models outperformed their SB-FCNN and MB-FCNN counterparts. This indicates that CNN models are better suited for fishing-vessel operational types recognition than FCNN models. The success of CNN models may be due to their ability to mitigate the impact of noise between neighboring trajectory points, which can arise from becalmed scenarios or missed reports in certain trajectory segments. By using a convolution kernel of a specific size to extract features from trajectory segments, CNN models effectively reduce the influence of such noise, unlike FCNN models, which are more susceptible to noise-induced classification accuracy reductions. Thus, the results confirm that 1D-convolution is superior to fully connected networks for feature extraction in sequential data.

## Effect of single- and multi-branch structures on model performance

As previously illustrated in Fig. 1, two distinct model-building strategies have been developed for 1D-sequence data: single- and multi-branch strategies. In the single-branch approach, multiple factors are concatenated and fed into the model sequentially as

depicted in Figs. 1A and 1B. In contrast, the multi-branch method involves inputting multiple factors into separate substructures, and then combining the extracted information before feeding it into the subsequent network for learning as depicted in Figs. 1C and 1D.

The experimental results presented in Table 3 indicate that, for both the FCNN and CNN structures, the multi-branch-based models exhibit higher classification accuracy than the single-branch models. Specifically, the MB-1dCNN model demonstrated improvements of 0.0011 in weighted *Accuracy* and 0.0015 in weighted *F1* score compared to the SB-1dCNN model. Similarly, the MB-FCNN model showed enhancements of 0.0524 in weighted *Accuracy* and 0.0835 in weighted *F1* score in comparison to the SB-FCNN model. This difference may be due to the number of neural networks used to handle feature extraction. The multi-branch structure allows for separate feature extraction and fusion of inputs, avoiding the noise impact that occurs when inputs are directly concatenated in the single-branch structure. Consequently, when dealing with multiple types of different inputs, a multi-branch structure can more effectively extract the features of the inputs and enhance the model classification accuracy. Overall, this outcome further demonstrates the advantages of the proposed MB-1dCNN model in identifying fishing vessel operational types.

## CONCLUSIONS

Accurate identification of the activity patterns or operational types of commercial fishing vessels is essential for the conservation and sustainable development of fishery resources. In this study, the MB-1dCNN model was developed to identify fishing vessel operation types using vessel trajectory data. The MB-1dCNN model is an end-to-end one-stage deep learning model that optimizes the overall objective of a task during the training process, and reduces the reliance on feature engineering common in traditional methods. Consequently, it reduces human intervention and is more suitable for the analysis of feature importance than existing alternatives. The Geohash geocoding algorithm was used to convert the geographic location of a track point into a literal representation and embedding techniques were used to extract spatio-temporal context information from trajectory sequences.

The MB-1dCNN model was compared with the SB-1dCNN, SB-FCNN, and MB-FCNN models in terms of recognition accuracy, and the proposed model exhibited better recognition performance for the three types of fishing operations considered in this study: trawl net, purse seine, and gillnet fishing. Overall, the 1D-CNN models exhibited better performance than the FCNN models when classifying the fishing vessels by operational type. Moreover, because multiple model input features were considered, both the 1D-CNN and FCNN models exhibited better performance under a multi-branch structure than a single-branch structure. Therefore, the combination of multi-branch structure and 1D-CNN has achieved superior results.

The influence of the Geohash length on the model accuracy was also explored. The weighted *Accuracy* and weighted *F1 score*s of the model tended to increase before

decreasing with increasing Geohash length. The highest classification accuracy was obtained when the Geohash length was 5, with a weighted *Accuracy* of 0.8595 ± 0.0082 and a weighted *F1* score of 0.8573 ± 0.0085. The spatial range represented by this Geohash encoding was approximately $4.89 \times 4.89$ km$^2$.

Overall, the MB-1dCNN model proposed in this article effectively realized fishing-vessel operational-type recognition by using the Geohash and embedding techniques and employing a convolutional neural network. Consequently, it reduces human intervention and is more suitable for the analysis of feature importance than existing alternatives. In the future, the authors will consider the importance or contribution of different features for the recognition effect using this method, and will compare this method with other methods such as LSTM to screen out a better model. Because vessels of different fishing operational types have different trajectory patterns in temporal and spatial distribution, better ways to obtain or identify these patterns will be explored in the future, possibly considering transformer models or graph neural networks.

## ACKNOWLEDGEMENTS

The authors thank the Editor, the Associate Editor, and anonymous reviewers for their valuable comments and constructive suggestions.

### Funding

This work was supported by the National Key R&D Program of China (2023YFD2401303) and the Central Public-Interest Scientific Institution Basal Research Fund, ECSFR CAFS (2022ZD0402). The funders had no role in study design, data collection and analysis, decision to publish, or preparation of the manuscript.

### Grant Disclosures

The following grant information was disclosed by the authors:
National Key R&D Program of China: 2023YFD2401303.
Central Public-Interest Scientific Institution Basal Research Fund, ECSFR CAFS: 2022ZD0402.

### Competing Interests

The authors declare that they have no competing interests.

### Author Contributions

- Bohui Jiang conceived and designed the experiments, performed the experiments, analyzed the data, performed the computation work, prepared figures and/or tables, and approved the final draft.
- Weifeng Zhou conceived and designed the experiments, analyzed the data, authored or reviewed drafts of the article, and approved the final draft.

## Data Availability

The data is available at figshare: Zhou, Weifeng (2024). trajectory data used in Peerj. figshare. Journal contribution. https://doi.org/10.6084/m9.figshare.27512454.v1.

The code is available in the Supplemental File.

## Supplemental Information

Supplemental information for this article can be found online at http://dx.doi.org/10.7717/peerj-cs.3020#supplemental-information.

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
