# Peer review of "Fishing operation type recognition based on multi-branch convolutional neural network using trajectory data"

_PeerJ Computer Science, doi:10.7717/peerj-cs.3020_

## Round 0.1 · original submission · Major Revisions

You have received 2 detailed reviews for your paper.

The reviewers raise several critical concerns:

1) The introduction needs to be rewritten to make the motivation and contribution clearer (Reviewer 1, 2)
2) The results need to be compared to a valid baseline (preferably one that does not use ANNs) , along with an LSTM model (Reviewer 1)
3) The architecture of the DNN needs to be shown in a figure (Reviewer 2)
4) The use of the embeddings and therefore the contribution of the paper is unclear (Reviewer 1, 2) – this is a critical point that needs to be properly addressed.

Due to the nature of the comments, I therefore suggest a risky major revision. The paper has a high chance of being rejected if the comments are not properly addressed.
Apart from the comments raised by the reviewers, I have the following comments:

1) The paper does not look very neat – the section headers are not large and bold and look like they are missing
2) The equations do not look very nice (especially equation 5) and this unfortunately affects readability and understanding.
3) I agree with Reviewer 1 that some of the equations (like that for accuracy, F1, etc.) can be avoided as it is basic textbook material!
I suggest rewriting the paper in latex and getting it edited by a professional English editor. Also, unless necessary, you do not need to cite the references provided by the reviewers.

Reviewer 1 ·

Basic reporting

The paper uses a good, readable English. Still, the the presentation becomes rather technical with little explanation of many concepts used. Sometimes things are taken for granted, like in line 272 where you use the term "input feature vector". What is the "input feature vector"?

The paper uses the term "word embeddings" a lot, although the data are not using words in the standard sense. It would have been better to apply only "embeddings" as what they are using is an embedding layer that creates embedding for positions, not words. The idea of (word) embeddings comes from Mikolov et al. back in 2013 and also other work from back then. The way it is written here seems to give credit to Zhao et al and Sonbol et al. for the concept.

In lines 106 to 111 there is talk about thresholds and features. Which thresholds/features are that? Later it occurs that your methods are also dependent on thresholds and features, like input data size, embedding size, layer sizes. Why are those not so critical as the ones mentioned in lines 106-111.

In lines 316 to 318 you indicate that you use mainly softmax as activation functions, and ReLU at the output layer. Is this correct? That is, in case, completely opposite of practice elsewhere. In case you need to explain. I guess it is just a writing error.

On lines 357-392 you give a one page description of evaluation metrics, which is basically textbook material. A much shorter discussion of those should be sufficient. And you could remove table 3. You could instead mention the use of performance metrics for multiclass prediction, including multiclass confusion matrices.You seem to be using some kind of weighting of results in table 4. Explain the weighting!

line 196: "We do" - not "we does".

in line 255 and 291 the use of "hat" above the y, makes the y too tiny compered to the fonts of the surrounding text. Fix!

Line 441: Use "reduced" instead of "avoided"

In line 444 you use the term "fully connected convolution". Writing error again?

The motivation for the study is solid. There is a need for good ways to analyse fisheries data. However, there is no good argument as to why it should be better to replace a location represented as two real numbers (latitude,longitude) with an embedding (which is a much longer real numbered vector).

The references below may be of interest and could be mentioned, although not addressing exactly the same problem. They also address geolocated fisheries data.

A. Ashrafi, B. Tessem, and K. Enberg, “Detection of fishing activities
from vessel trajectories,” in International Conference on Research
Challenges in Information Science. Lecture Notes in Business Information Processing (LNBIP, volume 514) Springer, 2023, pp. 105–120.

A. Ashrafi, B. Tessem, and K. Enberg, "Analysing Unlabeled Data with Randomness and Noise: The Case of Fishery Catch Reports", 14th Scandinavian Conference on Artificial Intelligence SCAI 2024, https://doi.org/10.3384/ecp208023


Other references seem to be ok, but the reference list is unordered and due to that it is hard to search for the references in the list. It is also formated a bit messy here and there, and maybe missing page numbers some places. Niemeyer's geohash method should have a reference to Niemeyer's own publications.

Figures are ok, but in Table 1 the 800 in the input layer and "20000,50" in the embedding layers should have been explained with relation to the original 5 data features. You could also explain what the square root sign in that table means.

Experimental design

The experiments are performed on real world data from China, available on a web site. It is good to have such prepared data available for the domain.

You have a nice figure indicating the distribution of the different data (Figure 2). In figure 2d it seems like the heading of the boat is very often close to 0.0, i.e., towards north. So we probably have missing/incorrect data here. It is not reasonable that the vessel heads north more often than any other direction. This could have been discussed, and if values are incorrect, how do you handle this.

The design of the experiments seems to follow standard machine learning practice.

I miss a comparison with a base line method that does not use the embeddings approach and even a non-deep learning approach. A main contribution of this paper is the replacement of original features with embeddings. If you cannot show an improvement from using a straight forward approach without embeddings your work has less value. This is a main flaw with the paper. This needs to be fixed! You could use a lstm model, with cyclical features for heading and point in time, and/or short sequences of tracking points combined with a non-deep learning method.

Validity of the findings

The methods presented are not compared with a base line not using embedding, so we do not really know the quality of the results. 85% accuracy does not seem to be very high if you compare with ML experiments in other domains.

In line 141-142 there is a claim that the findings will be a key reference in the field. This is a very strong claim. I do not accept this unless you compare with other approaches showing that the approach in fact give better results.

You are using embedding layers in all models, which may be smart. However, this will create completely different embeddings for each model and feature. This needs to be discussed. You also need to verify that the embeddings created in fact are meaningful. For example, close geographical points should most often have close embeddings in embedding space, unless for example there is some kind of particular fishing type going on at a very restricted location/time. Then two neighboring geolocations may have different semantics.

A couple of other relevant questions to be addresed:
Should you use one common embedding layer for all the 5 models?
Is the data set big enough to be able to get good enough embeddings?

Additional comments

The work is well motivated at a general level, i.e., we need methods for analysing fisheries data. However, the use of embeddings to represent features combined with CNN-s is not really confirmed to be a superior method. Some additional work needs to be done to investigate the value of the described methods. In addition, there are presentation issues to be fixed in what I consider a fairly readable text.

Reviewer 2 ·

Basic reporting

- the article needs substantial improvement in terms of language and professional English. It has many grammatical and punctuation errors and needs to be read by native English speaker.

For example, authors say "We does not directly use functions from the Gensim library," it should be "We do not directly,"
also, authors say "All these methods exhibited good performance as regards", this is emotional and the authors should use more professional English terms. For example "All these methods exhibited plausible performance as regards..."

- in terms of literature, The article does not include sufficient introduction and background to demonstrate how the work fits into the broader field of knowledge. The introduction does not state clearly the contributions of the authors and the work. In the introduction, the problem description is very short and unclear. It needs to be expanded and detailed explanation should be rather provided.

also, in the introduction, authors say "The insights garnered from this study are valuable for industry applications and establish it as a key reference in the field." the claims are ambitious and high but no evidence is discussed. The authors should provide evidence about those claims and how their work provides substantial contribution to the body of the knowledge.


in terms of the Professional article structure, figures, tables. Raw data shared. I believe that the literature review and state-of-art section should be placed separately in a dedicated section, as the current version only list insufficient literature as part of the introduction and this is irrelevant and should be expanded to include other related methods from the literature.
Also, the raw data discussed in the paper which the authors have used for testing is not publicly accessible and hard to access in order to evaluate the models presented by the authors. The data should be made available publicly on a public repository on GitHub repository for example in order for the author to check the validity of the proposed models.
the provided link by authors:
https://tianchi.aliyun.com/competition/entrance/231768/introduction
even after registration to Alibaba, does not provide access to data!
appropriate raw data should have been made available in accordance with Data Sharing policy of PeerJ journals.


For figures, Authors should provide figures explaining how geohash works. Refer to examples available in the suggested literature list above.
also, the geohash references provided by authors are incorrect. Authors say "The geohash algorithm (Yin et al., 2022; Zhou et al., 2021)," those are incorrect references of geohash.
Maybe, you can use one of the references provided above to explain geohash in your paper.

Experimental design

in the experimental design, Research question is ill-defined, irrelevant and need to be restated clearly. The authors need to clearly define the problem statement, probably also providing a subsection showing a clear scenario with dummy figures showing exactly what was the problem and how it is solved by the proposed model. Also, It is advisable to add a research problem statement section that clearly describes the problem statement, probably with examples and mathematically-principled statements that shows the definition formally of all elements of the model including those that are in-between stages of running of the system.
The problem statement and the design of the framework should clearly state how research presented in this paper fills an identified knowledge gap, which i really found that is hard to read and search within the text of the current version of the manuscript

in terms of the fact that the investigation must have been conducted rigorously and to a high technical standard, the recent study is proposing a method that has been applied only to one dataset, with very few metrics of evaluation. Also, it does not compare properly to any other similar work from the recent state-of-art. Having said that, the authors need to perform more testing with more than one dataset, and compare with a very clear related similar system from the literature, in order to validate the proposal presented in the paper and the claims therein.

Methods are not sufficiently described with information to be reproducible by another investigator. As we stated previously, the data should be made publicly accessible.

the incorporation of the convolutional neural network (CNN) model with spatiotemporal contextual information is not clearly defined and needs to be expanded. It is unclear how the spatiotemporal contextual information was incorporated within the CNN model. There are several types of such kind of contextual incorporation, including pre-filtering and post-filtering approaches. They are clearly discussed in a very relevant work from the recent related state-of-art, which can be also used in this paper and cited appropriately. the suggested reference is:
- Al Jawarneh, I. M., Bellavista, P., Corradi, A., Foschini, L., Montanari, R., Berrocal, J., & Murillo, J. M. (2020). A pre-filtering approach for incorporating contextual information into deep learning based recommender systems. IEEE Access, 8, 40485-40498.

Also, the work in this paper should compare to the work presented in the reference suggested as they are very much related in terms of the need to incorporate contextual information into deep-learning based models to enhance predictability and recommendations.

Validity of the findings

The data on which the conclusions are based were not provided or made available in an acceptable discipline-specific repository. The provided link does not allow public access to the data
https://tianchi.aliyun.com/competition/entrance/231768/introduction),
even after registering to Alibaba Cloud as suggested, it does not allow public access to the data.

The data must be made publicly available, otherwise the study is useless, as the claims are hard to be validated and reproduced for comparison.

The conclusions are not appropriately stated and THE CONCLUSION section should be expanded greatly to include future research perspectives and the conclusion drawn from the results should be discussed in more details , and should be connected to the original question investigated, and should be limited to those supported by the results.

Additional comments

- there should be an architecture overview section, showing a very clear figure depicting the system components, as a pipeline probably from left to right, explaining in deep details the mechanism by which the system generally operates and the workflow, in addition to the responsibilities of each component, in addition to a clear explanation of how those components work synergistically to achieve the overall goals of the system design.

---

## Round 0.2 · Major Revisions

Here's a condensed meta-review combining the reviewers' comments:

1. Language and Structure (Reviewer 1, 2):
- Improve English grammar, word choice, and readability
- Enhance explanation of tables and figures
- Strengthen the literature review section

2. Methodology Clarity (Reviewer 1, 2):
- Clarify input data organization and processing
- Better explain the model architecture, especially the 1D convolutional layer and multi-branch structure
- Elaborate on the embedding approach and its rationale

3. Experimental Design and Validation (Reviewer 1, 2):
- Include comparisons with non-embedding CNNs and alternative approaches
- Test with multiple datasets
- Provide more robust justifications for methodological choices

4. Result Interpretation and Reporting (Reviewer 1):
- Improve explanation of zero values in direction data
- Demonstrate the usefulness of the 50-dimensional vector representation

5. Data Accessibility (Reviewer 2):
- Make dataset easily accessible through open repositories like GitHub or Zenodo

6. Problem Statement (Reviewer 2):
- Add a clear "Research Problem Statement" section
- Better define how the research fills an identified knowledge gap

Please include a document where you address these points as part of your revision.

Reviewer 1 ·

Basic reporting

The English is fairly ok, but with omitted words, somewhat strange choice of words some places. The language could still be improved. The claims of the paper contribution is repeated too may times.

Structure is fine, figures are nice, but Table 1 is not explained sufficiently. What does 20000,50 mean? What does 5,4 mean? I do not find any explanations about this in the text or in the caption. Input has the value 800. What does this number mean?

The results seem to verify that the MB-1dCNN seem to be the best solution among those tested. However, there is no comparison to a CNN not using embeddings. How can you claim that the use of embedding is an improvement when you do not compare with alternatives, for example using latitude, longitude directly or using Space2Vec or similar techniques. So far, I am not convinced.

The organisation of the input data is not clear. Each data point consists of a sequence of vessel data (a time series), each element in the sequence contains: a location, a time stamp, speed and heading. locations are converted into a 50-dimensional vector (50 is my assumption, it is not explained anywhere), and so are time stamps. This gives every element a 102-dimensional representation before sent to a 1-d convolutional layer. 1-d convolution is based on on the use of one-dimensional vectors. How is this 1-d vector built from the series of 102-sized elements? It is no at all clear to me. In fact it looks like it is 2-dimensional.

In the explanation of the multi-branch structure the authors write about "factors". What are factors in this context? It is not explained at all.

In the rebuttal the authors suggest that the amount of zero-es in the data on direction could be because vessels turn north due to currents/weather or similar. This is not a good explanation. 0 is north, but so is 360, so if the authors suggested explanation is correct, then there should be more data also in the interval from 350 to 360 (for example).

In general I find that the paper lacks quite a bit in terms of explaining the method, and also in terms of usefulness of the approach.

Experimental design

The experimental design is not entirely clear as the organisation of the input data is not well explained. See other comments!

Accuracy results and precision/recall are reported and give insight into the results.

Validity of the findings

I do not find the the results convincing. The authors basically exchange 2 numbers (longitude/latitude) with 50(?) numbers. It is not verified that this actually is useful in this context. In general a CNN without the embeddings should be enough to do this. Since we have no comparison it is hard to judge. In the rebuttal the authors refer to some other work not yet published, but this is not sufficient for validating the value of the work.

Reviewer 2 ·

Basic reporting

still I can find grammatical mistakes and the readability can be further improved. For example authors say "The research motivation of the study was to build a better neural network model and to improve the recognition effect of fishing vessel operation-types, reducing the dependence of previous methods on feature engineering and making the most of spatiotemporal contextual information. ". Do not use past tense, use present tense as possible. For example "The motivation of the study is to build ...."

Also, you say "and making the most of spatiotemporal contextual information" --> you can instead say "and exploiting extra spatiotemporal contextual information....", use more formal language however possible.

Other mistakes in the introduction, such as you say "spatioltemporal contextual feature extraction of trajectory sequences." --> replace "spatioltemporal " with "spatiotemporal"

And again, avoid unmeasurable facts, so you say "to build a better neural network model". how "better"? So, you can say "to design a new neural network model ....."

Also, "This mothod was applied in experiments to identify fishing" --> change to "method"

in the "materials and methods" section, you say "This section provides the descriptions " --> remove "the"

you also say "All the architectures can be found in Figure 1" --> remove "the",,,,
avoid using "the" a lot!

also, in comment 4, i have suggested that the authors dedicate a section for "literature review". I still believe that the literature review provided by the authors is weak and need to be improved. Specifically, discussing similar techniques from the literature and comparing and finding gaps and reflecting on how the methods presented in the paper are closing those gaps.

Experimental design

Authors said "so the structure of SB-FCNN in Figure 1 can be thought as a kind of the architecture of the DNN". I disagree with the authors on this. The architecture should clearly show how their additions incorporate deeply into the baseline models atop which their system is built.

Validity of the findings

the authors response to my comment 5 about the public availability of the data used for experimentation.

"The provided link should not require registration, but it is a Chinese website and may have some obstacles for global visitors. The reason why only some of the data is uploaded is because the Peerj website has a limit on the amount of data that can be uploaded."

I tried several times downloading the data, and each time i get the message "You need to sign up for the game to download data set!".

I believe the authors should upload the data to open repositories such as GitHub, or Zenodo.

Additional comments

Authors should provide figures explaining how geohash works, draw an illustrative figure showing how geohash encoding works in terms of the coverage. How geohash bounding boxes can cover the study area!

to my comment "[7] in the experimental design, Research question is ill-defined, irrelevant and need to be restated clearly. The authors need to clearly define the problem statement, probably also providing a subsection showing a clear scenario with dummy figures showing exactly what was the problem and how it is solved by the proposed model. Also, It is advisable to add a research problem statement section that clearly describes the problem statement, probably with examples and mathematically-principled statements that shows the definition formally of all elements of the model including those that are in-between stages of running of the system."
the answer provided by authors is not compelling! Authors say "-On [7]:The first and second paragraphs of the introduction describe the application scenarios of the problem. In other literature read by the author, there is generally no research problem statement section, and the research opinions and application scenarios are explained in the introduction. Compared to other application scenarios of deep learning, the recognition of fishing vessel operation types is undoubtedly a very narrow field."
I still believe that the problem is ill-defined and need to be restated clearly!

Also, the authors answer to my comment "[8]The problem statement and the design of the framework should clearly state how research presented in this paper fills an identified knowledge gap, which i really found that is hard to read and search within the text of the current version of the manuscript" is not compelling, still the problem addressed is unclear and the paper is hard to read.

There are a lot of trajectory data, and I still believe that the "authors need to perform more testing with more than one dataset, and compare with a very clear related similar system from the literature, in order to validate the proposal presented in the paper and the claims therein.". The answer of the authors to this concern is not compelling

---

## Round 0.3 · Major Revisions

You have received one review for your revision and the reviewer suggests minor revision.
Apart from the comments of the reviewer, I have the following suggestions:

1) the Abstract is too long, and needs to be shortened.

2) there are many typos in the paper. Just in the Introduction the 2nd line has "oceans as the major parts of the Earth, which cover more than 70%thís surface" - should be "oceans, which cover more than 70% of the surface". Please go through the paper thoroughly and employ the services of a professional editor.

3) The related work section is too small, you need to perform a more structured literature review - please see Webster and Watson, MISQ 2002.

4) The research research problem statement needs to be merged with the Materials and Methods (or Research design) section.

5) The mathematical equations (1, 2, 3) are not well formatted
- the equations 4, 5 are even worse.

6) the reasoning given for the using the NN are insufficient: what were the reasons why the NNs in Figure 1 were considered in this study? This is not clear at all!

7) the presentation and explanation of the results could be improved (also by following the advice of the reviewer).

**Language Note:** The Academic Editor has identified that the English language must be improved. PeerJ can provide language editing services - please contact us at [email protected] for pricing (be sure to provide your manuscript number and title). Alternatively, you should make your own arrangements to improve the language quality and provide details in your response letter. – PeerJ Staff

Reviewer 2 ·

Basic reporting

The manuscript generally meets the standards for basic reporting. The English is clear and professional, though minor grammatical refinements could enhance clarity in a few instances (e.g., lines 23, 77). Literature references are adequate, providing sufficient context for the study, but expanding on recent advancements in spatiotemporal data processing could strengthen the background. The article structure aligns with professional norms, and figures/tables are well-labeled and relevant. Raw data has been shared, which is commendable. For improvement, briefly define key terms like "geohash" earlier in the text for broader accessibility. No major issues were found.

Experimental design

The manuscript presents original research within the journal's scope, addressing a relevant and meaningful research question related to fishing vessel operation type recognition. The study effectively identifies a knowledge gap in utilizing spatiotemporal contextual information for trajectory data analysis. However, the methods section could benefit from additional detail regarding the preprocessing steps of the geohash encoding and embedding techniques to ensure full replicability. Specifically, providing more insight into parameter tuning and data normalization processes would strengthen the technical rigor. No major issues were identified. Minor revision recommended.

Validity of the findings

The findings are valid and well-supported by the provided data, which is robust and statistically sound. The authors have supplied all underlying data, enabling meaningful replication. However, the manuscript could strengthen its rationale for the chosen experimental setup and further clarify the benefits of the MB-1dCNN model compared to existing methods in terms of practical application. While the conclusions are appropriately linked to the research question, a more detailed discussion on the limitations of the study would enhance transparency. Minor revisions are recommended to address these aspects.

---

## Round 0.4 · Minor Revisions

We invite the authors to address the reviewer’s comments, particularly regarding the clarity in the description of parameters, the reformulation of overly strong statements, and the discussion of the limitations related to the data used. As the comments are specific and limited in scope, we ask the authors to make the necessary revisions to fully satisfy the reviewer.

Reviewer 1 ·

Basic reporting

This paper is readable and well organized. It has the conventional structure of a traditional applied machine learning paper. Results are well organised and presented.

The presentation since I read the first version has improved much as the authors now are able to convince me that it may be useful to use geohash embeddings to represent locations at sea.

In line 250 it would have been even more clear if you wrote
"The parameters are set to 20,000 (vocabulary size) and 50 (embedding dimension)". It is not so that every reader knows the parameters of TensorFlow components.

I am not a particular fan of accuracy numbers etc. in abstracts. They are basically uninteresting as part of an abstract. Why not just say that a particular method outperforms the other in terms of accuracy and F1-scores.

Experimental design

No comments

Validity of the findings

The idea of geohash embeddings gives interesting results and can encourage other researchers to explore the same technique for other problems, data sets and deep learning architectures. For me this is more important than the various architectures tested here, but the experiments indicate that architecture is also very important.

I would not, however, claim that it is "crucial" to use such embeddings, like the authors do on line 407-408. It "promises to play a significant role" would be a better statement. You cannot say that this approach will not be out-competed by new ideas.

The heading/direction variable has a skewed distribution, with very many close to 0. This indicates that there is incorrect reporting of this feature. The argument (as written in one of the rebuttals from the authors) that 0 is the same north is not valid, as 359 is also close to north. It probably has nothing to say for the learned ML models, as they may learn to ignore the feature. But maybe removing that feature could give better results.

The data are from only three vessels (should be mentioned), which makes the models not so interesting in a real world context. The particularities, reporting practices, and preferences of a particular vessel skipper may have more impact on the model, perhaps even more than locations and speed. For example if one of the vessels (let us say the one that does gill nets) does not report true heading values, but just sets this to 0, then "gill nets" may be learned from reporting 0 as heading instead of the movement pattern. I guess much more (verified) data is needed to train an operational model. The authors should comment on this in the discussion.

Additional comments

There is some strange use of hyphens, particularly in the early part of the text, like writing "machine - learning" instead of "machine learning".

---

## Round 0.5 · accepted · Accept

I thank the authors for the work carried out in revising the manuscript. All previous comments and observations have been addressed clearly and thoroughly. The changes made have improved the overall quality of the contribution, which is now suitable for publication.